# Therapeutic Drug Monitoring of Anti-TNFα Inhibitors: A Matter of Cut-Off Ranges

**DOI:** 10.3390/pharmaceutics15071834

**Published:** 2023-06-27

**Authors:** Stefania Cheli, Diego Savino, Francesca Penagini, Gianvincenzo Zuccotti, Giovanna Zuin, Emilio Clementi, Dario Cattaneo

**Affiliations:** 1Unit of Clinical Pharmacology, ASST Fatebenefratelli Sacco, University Hospital, 20157 Milano, Italy; diego.savino@studenti.unimi.it (D.S.); dario.cattaneo@asst-fbf-sacco.it (D.C.); 2Pediatric Department, “Vittore Buzzi” Children’s Hospital, University of Milan, 20154 Milan, Italy; francesca.penagini@asst-fbf-sacco.it (F.P.); gianvincenzo.zuccotti@unimi.it (G.Z.); 3Pediatrics, IRCCS San Gerardo dei Tintori Foundation, 20900 Monza, Italy; giovanna.zuin11@gmail.com; 4Scientific Institute IRCCS Eugenio Medea, 23842 Bosisio Parini, Italy; emilio.clementi@unimi.it; 5Clinical Pharmacology Unit, Department of Biomedical and Clinical Sciences, L. Sacco University Hospital, Università degli Studi di Milano, 20122 Milan, Italy

**Keywords:** therapeutic drug monitoring, inflammatory bowel disease, infliximab, adalimumab, anti-infliximab antibodies, adalimumab antibodies

## Abstract

Therapeutic drug monitoring (TDM) is a useful tool for optimising the use of anti-TNFα inhibitors in patients with inflammatory bowel diseases (IBDs). Recently, point-of-care methods for the quantification of drug levels and anti-drug antibodies (ADAs) have been developed to overcome the limitations of conventional enzyme-linked immunoabsorbent assays (ELISAs). Here, we evaluated the performance, interchangeability, and agreement between an automated ELISA-based immunoassay (CHORUS Promonitor) and the lateral flow assay (RIDA^®^QUICK) for the quantification of infliximab (IFX, *n* = 65) and adalimumab (ADM, *n* = 58) plasma levels in IBD patients. Thirty-two samples for IFX and twenty-three samples for ADM that tested positively for the presence of ADAs were also used. Overall, data analysis showed a good agreement of ADM trough concentrations (R^2^ = 0.75) between the two assays as well as for ADA measurement (K > 0.8). However, IFX levels highlighted a weak correlation (R^2^ = 0.58) between the two kits, with the RIDA^®^QUICK assay overestimating IFX plasma values by 30% when compared to the CHORUS Promonitor kit. Results from this study show that the two assays are not quantitatively and qualitatively interchangeable due to substantial discrepancies in some results. Accordingly, the same assay should be used for the longitudinal follow-up of IBD patients.

## 1. Background

The response of patients with inflammatory bowel diseases (IBDs) to anti-tumor necrosis factor alpha (TNFα) agents (infliximab, adalimumab, golimumab, and certolizumab), in terms of the durability of drug therapy, reduction of antibody formation risk, severe infusion reactions, and decrease in hospitalisations, can be improved by the therapeutic drug monitoring (TDM) of blood drug concentrations [1,2]. In children and adolescents, infliximab (IFX) and adalimumab (ADM) are currently the only biotherapies approved by the Food and Drug Administration (FDA) and the European Medicines Agency (EMA) for the treatment of paediatric IBD [3]. Increasing evidence in children supports the update of European guidelines on optimising the treatment of IBD by early proactive TDM followed by dose optimisation in patients on anti-TNF agents [4]. While IFX and ADM have a significant therapeutic impact in both paediatric and adult populations, a small yet clinically relevant proportion of IBD patients, between 10% and 25%, fail to respond to the initial treatment. These patients are referred to as primary anti-TNFα non-responders [5,6]. Subsequently, secondary loss of response may also be observed, which could be driven by the development of drug-specific antibodies (ADAs) owing to, in some individuals, anti-TNFα inhibitors possibly inducing immunogenicity with possible loss of efficacy and delayed-type hypersensitivity [7,8]. Up to 73% of patients receiving IFX and up to 35% of patients receiving ADM report the formation of persistent ADAs [9]. About one-third of these patients usually experience a loss of response because of developing ADAs, which usually occurs within 12 months after the start of treatment [9]. Therefore, to improve the management of patients with IBD, anti-TNFα drugs’ trough level and ADA determination is commonly carried out (so-called proactive TDM) [9,10]. Numerous exposure–outcome relationship data from prospective studies and post hoc analyses of randomised controlled trials (RCTs) have shown the benefits of proactive TDM over empiric or reactive dosing [1,11,12,13,14]. Proactive TDM can also have an important role when de-escalating therapy, proving to be more cost-effective [1,11,12,13,14].

TDM has been the most promising approach for treating secondary anti-TNF drug failure since it has been associated with improved response rates [10,11,12,13,14]. Several assays are available on the market to measure IFX and ADM trough levels and related anti-drug antibodies (ADAs) [15]; of these, the enzyme-linked immunosorbent assay (ELISA), the radioimmunoassay (RIA), and the homogenous mobility shift assay (HMSA) are the three most commonly used [15,16]. Currently, the solid phase assay (ELISA) is the only technique that can be used for the TDM of all TNFα inhibitors as well as for the quantitative investigation of ADAs, resulting in the gold standard technology. The main benefits of an ELISA are its user friendliness and the commercially available ready-to-use kits. The fluid phase systems (RIA, HMSA) are well known for their ability to identify low-affinity and monospecific ADAs but are indeed less frequently implemented in clinical practice since they are more demanding techniques. These methods are time-consuming and require sophisticated equipment, reagents, and technical expertise. The development of a technique based on rapid point-of-care tests (POCTs) in lateral flow immunochromatography, however, has simplified the methodology to measure plasma IFX and ADM concentrations. This provides advantages such as the possibility to test individual samples, comparatively simple operation, and quick turnaround for results (15–20 min). The IFX and ADM trough levels have been compared in numerous studies, showing overall good correlations between the methods [16,17,18,19,20]. However, several studies reported differences in the absolute drug concentrations, and this can impact clinical practice when thresholds or ranges of concentration are the target to reach by TDM [18,19,20]. Regarding ADA monitoring assays, associations have been reported, with correlation coefficients ranging from 0.54 to 0.99 [17,18,21]. Nevertheless, the different capacities of available assays to detect ADAs in the presence of the drug can lead to differences in sensitivity [22]. Conventional immunoassays, such ELISA, are routinely used for ADA screening, although they have a low drug tolerance and cannot distinguish neutralising and non-neutralising ADAs [22]. Here, we compared a POCT, an ELISA-based method for IFX and ADM trough level quantification, and their ADAs in a small cohort of patients with IBD. The main goal of this comparative study was to evaluate the agreement between these approaches for the currently accepted therapeutic ranges [22,23] as well as their interchangeability.

## 2. Materials and Methods

### 2.1. Laboratory Assays

Plasma trough concentrations of IFX and ADM were measured with two different commercial kits: RIDA^®^QUICK Monitoring (R-Biopharm AG, Darmstadt, Germany) and CHORUS Promonitor (Diesse Diagnostica Senese, Siena, Italy) according to the manufacturers’ instructions. The POCT-based lateral flow assay RIDA^®^QUICK Monitoring uses a chip card to transmit test data and a calibration curve to the reader. Following centrifugation, plasma samples were diluted with buffer (1:50 or 1:200) and loaded into the well of the test cassette, in accordance with the manufacturer’s instructions. After 15 min, the results are displayed on a screen. The detection limit is less than 0.5 µg/mL for IFX and 0.32 µg/mL for ADM, according to the manufacturer. CHORUS Promonitor is a capture (IFX kit) or sandwich (ADM kit) ELISA, with microwell strips precoated with an anti-TNFα human monoclonal antibody bound to human recombinant TNFα. The CHORUS Trio instrument is used to assess the intensity of the color produced after adding a chromogenic substrate to determine the enzyme activity (Diesse Diagnostica Senese, Siena, Italy). The disposable devices contain all the reagents to perform the test when applied to this instrument. The results are expressed in µg/mL calculated on a batch-dependent curve stored in the instrument, and the detection limit is less than 0.3 µg/mL both for IFX and ADM.

The search for anti-infliximab and anti-adalimumab antibodies was carried out using ELISA techniques supplied by the same manufacturers: RIDASCREEN^®^ Anti-IFX/ADM Antibodies (R-Biopharm AG, Darmstadt, Germany) and CHORUS Promonitor Anti-Infliximab/Adalimumab (Diesse Diagnostica Senese, Siena, Italy). All tests were performed following the manufacturers’ instructions. All evaluated tests use a bridging assay strategy to capture free ADAs. This strategy takes advantage of ADA bridging properties, which are bivalent antibodies. The main assay differences regard the units and immune complex revelation steps: RIDASCREEN^®^ expresses ADA results in ng/mL whereas CHORUS Promonitor gives results in arbitrary units; the other difference is the revelation strategy, which is a two-step biotin/streptavidin revelation for RIDASCREEN^®^, as opposed to CHORUS Promonitor, which is a one-step procedure. Controls provided by manufacturers for each assay were processed before every analytical series, and their values were checked to ensure they were within the expected range. The interpretation of the results was carried out according to the cut-off provided by the manufacturers.

### 2.2. Cut-Off Ranges

According to the manufacturer’s instructions for the RIDA^®^QUICK Monitoring assays, a therapeutic trough concentration window of 3–7 mg/L is recommended for IFX, following the TAXIT algorithm [11], while a target range of 5–10 mg/L is recommended for ADM [10]. There is no cut-off proposed for the CHORUS Promonitor assay result interpretation, but the test on the examined plasma can be interpreted as positive when the result is ≥0.30 mg/L and negative when the result is <0.30 mg/L. There is no concentration threshold for CHORUS Promonitor assays associated with clinical significance. In addition, for ADA result interpretation, there is no cut-off proposed in the manufacturers’ instructions for RIDASCREEN^®^ and CHORUS Promonitor kits. Therefore, ADAs are considered positive only when they are detectable.

### 2.3. Sample Selection

The samples for this study were collected from paediatric and adult patients with IBD who received treatment with IFX or ADM and underwent routine determination of plasma concentrations and ADA levels at the Unit of Clinical Pharmacology of the Luigi Sacco University Hospital (Milan, Italy). All patients with an established diagnosis of Crohn’s disease (CD) or ulcerative colitis (UC) were included. Plasma samples were collected at the trough level both during the induction and maintenance phases. Samples were excluded if there was not sufficient plasma to run them through both assays and if they were hemolysed. In addition, patients were excluded if data on IFX or ADM dosing or levels were missing. A total of 123 (65 for IFX and 58 for ADM) plasma samples from 123 IBD patients treated with anti-TNFα inhibitors were tested for trough concentration measurement using RIDA^®^QUICK Monitoring and the CHORUS Promonitor test. Forty-two samples tested positive for the presence of ADAs (twenty-six for IFX and sixteen for ADM) by the RIDASCREEN^®^ assays, which were also analysed with the CHORUS Promonitor tests.

### 2.4. Data Analysis

The normality of IFX and ADM plasma concentration distributions for each method was checked using the Kolmogorov–Smirnov test. Descriptive analysis provided the median, 25th and 75th percentiles (IQR), and ranges of the different assays. Passing–Bablok regression, the Bland–Altman method of estimating bias, and the determination of Pearson’s correlation coefficient were performed for all comparative analyses. Bland–Altman plots were created to graphically assess agreement between the assays by accounting for the difference in absolute concentration [24]. In this graphical method, differences between the assays are plotted against the averages of the pair, and bias is then calculated. For qualitative comparison, patients were categorised by therapeutic interval (<3 mg/L, ≥3 to 7 mg/L and <7 mg/L for IFX concentrations, and <5 mg/L, ≥5 to 10 mg/L and <10 mg/L for ADM concentrations, according to the manufacturers’ instructions). The agreement between results obtained with the ADA detection kits, RIDASCREEN^®^ and CHORUS Promonitor kits, was assessed using Cohen’s kappa coefficient, which has a value of “0” if there is no more agreement between two tests than may be predicted by chance and “1” if there is perfect agreement [25] (Appendix A).

## 3. Results

### 3.1. Infliximab Trough Levels

The IFX plasma concentrations, measured in 65 IBD patients for each method, were first checked for normality, and data from the CHORUS Promonitor assay showed a non-normal distribution (*p*-value < 0.05). The median values were 9.8 mg/L (IQR: 4–15.8) for the RIDA^®^QUICK and 5.8 mg/L (IQR: 2.5–9.2) for the CHORUS Promonitor assay, with a significant difference between the two assays (*p*-value < 0.05) (Figure 1A). This trend was confirmed by linear regression showing poor correlation between the RIDA^®^QUICK and CHORUS Promonitor assays (R^2^ = 0.58). The slopes and intercept have also been calculated and are shown in Figure 1B; the linear correlation between the two assays does not consider the difference in absolute concentration, making it an unsuitable test to compare analysis methods. For this reason, we conducted a Bland–Altman analysis to complete the comparison and establish the level of agreement between the two assays. This study revealed a high level of systematic bias (−39.2%) between the two assays (Figure 1C).

Considering this finding and taking into account that statistically concordant data can lead to a discordant classification in therapeutic concentration ranges, we performed a qualitative analysis of the concordance of these two tests. According to the IFX therapeutic window, the patients were categorised into three groups, as shown in Table 1: 15 (23%) patients for the RIDA^®^QUICK assay vs. 22 (33.8%) for the CHORUS Promonitor assay in the subtherapeutic group (<3 mg/L); 10 (15.5%) patients for the RIDA^®^QUICK assay vs. 23 (35.4%) for the CHORUS Promonitor assay in the therapeutic range (3–7 mg/L); and 40 (61.5%) patients for the RIDA^®^QUICK assay vs. 20 (30.8%) for the CHORUS Promonitor assay in the supratherapeutic group (>7 mg/L).

Since the RIDASCREEN^®^ assay exhibited higher IFX concentrations than the CHORUS Promonitor one, the classification range for all these samples differed between the two assays, with 30% of these patients shifting to a higher category. Consequently, the strength of the agreement according to the kappa result (0.41 ± 0.078) was “moderate” between the RIDASCREEN^®^ and CHORUS Promonitor assays (Table 1).

### 3.2. Adalimumab Trough Levels

The plasma ADM trough levels of 58 IBD patients were measured. The results of the two assays show a normal distribution of ADM values. The median values were 9.1 mg/L (IQR: 3.5–15) for the RIDA^®^QUICK assay and 8.0 mg/L (IQR: 3.6–15.2) for the CHORUS Promonitor assay, without a significant difference between the two (*p* > 0.05) (Figure 2A). The two assays exhibited a linear quantitative correlation with a good coefficient of determination (R^2^ = 0.757) (Figure 2B). Bland–Altman analysis indicated that the two assays had good agreement, with a bias measured at −2.2% and only four values (3.4%) beyond the 95% limit of agreement (Figure 2C). The results for each assay were sorted into categories according to the therapeutic window for ADM concentrations.

Table 2 provides a summary view showing: 18 (31%) patients for the RIDA^®^QUICK assay vs. 17 (29.3%) for the CHORUS Promonitor assay in the subtherapeutic group (<5 mg/L); 16 (27.6%) patients for the RIDA^®^QUICK assay vs. 22 (38%) for the CHORUS Promonitor assay in the therapeutic range (5–10 mg/L); and 24 (41.4%) patients for the RIDA^®^QUICK assay vs. 19 (32.7%) for the CHORUS Promonitor assay in the supratherapeutic group (>10 mg/L). As a result, the agreement between the RIDASCREEN^®^ and CHORUS Promonitor assays was “substantial”, according to the kappa result (0.66 ± 0.082).

### 3.3. Anti-Infliximab and Anti-Adalimumab Antibodies

Thirty-two samples for IFX and twenty-three for ADM were collected with undetectable drug levels (<0.5 mg/L). All 55 samples were run on the RIDASCREEN^®^ and CHORUS Promonitor assays. We only conducted a qualitative comparison between the two assays due to variations in measurement units between the two suppliers (arbitrary units for the CHORUS Promonitor assay and ng/mL for the RIDASCREEN^®^ assay) and a small number of individuals developing ADAs (Table 3). One patient (3.1%) was found positive for ADAs to IFX only with the RIDASCREEN^®^ assay, but with an ADA level measured close to the lower detection limit of the method (2.83 ng/mL with a limit at 2.5 ng/mL), and negative with the CHORUS Promonitor assay (lower detection limit of 2 AU/mL). Nevertheless, perfect agreement between the two assays was revealed by an analysis using Cohen’s kappa (K = 0.904 ± 0.094) (Table 2). The two assays also had almost perfect agreement when determining the detection of ADAs to ADM, according to a pairwise analysis by Cohen’s kappa (K = 0.819 ± 0.120) (Table 3). Two patients (8.7%) were found positive for ADAs to ADM with the RIDASCREEN^®^ assay and negative with the CHORUS Promonitor assay, with one of them measuring close to the lower detection limit of the method (8.5 ng/mL with a limit at 5 ng/mL for the RIDASCREEN^®^ assay and <6 AU/mL for the CHORUS Promonitor assay). In the other patient with discordant results, the values obtained by the RIDASCREEN^®^ assay were remarkably high (above the positive detection threshold: >10.000 ng/mL) and lower than the threshold (<6 AU/mL) with the CHORUS Promonitor assay, despite appearing overexposed. It was possible to measure ADAs in this patient with the CHORUS Promonitor test after diluting the sample up to 1:10.000, with a result of 20.500 AU/mL.

## 4. Discussion

Measurement of anti-TNFα trough blood levels is considered a useful diagnostic approach to optimise the response in patients receiving anti-TNFα therapy, ideally when paired with the detection of ADAs [12,23]. To this end, an important prerequisite is the availability of analytical methods with rapid turnaround times that can concomitantly assess both concentrations and ADA levels of TNFα inhibitors. Consequently, POCTs have grown more widely used in laboratories since their introduction due to their accessibility, ease of use, and quickness. Despite this significance, a method comparison of POCT and ELISA techniques is required to evaluate whether these may be interchanged. This comparison had not been carried out to validate the recently recommended therapeutic ranges (based mostly on ELISAs). Current evidence suggests that despite a good mathematical correlation between different assays, including the commonly used ELISA, radioimmunoassay, HMSA, and POCT, they can lead to different therapeutic decisions [26,27].

This is the first study comparing the performance of the automatic immunometric assay, a monotest device based on ELISA (CHORUS Promonitor assay), for TDM of IFX and ADM with a commercially available POCT (RIDA^®^QUICK assays). Our results show that the comparison of the two methods for IFX trough levels is not satisfactory, with low agreement and high biases (up to 20%). The greatest discrepancies were observed at higher levels of IFX, resulting in a relevant discordance in classifying patients with supratherapeutic values. Each individual measurement is intrinsically associated with a measurement uncertainty due to coefficients of variation (inter-assay variability of the method). The POCT, as previously described in the literature, provides a systematically higher value of IFX trough levels than those determined by ELISAs [27,28,29], leading us to conclude that the target for IFX levels in TDM should differ depending on the measurement method. As seen in Table 1, only 60% of patients have therapeutic range agreement between the two assays, while the remaining 40% have supratherapeutic values with the RIDA^®^QUICK assay compared to the CHORUS Promonitor assay. Previous method comparison studies have demonstrated an overestimation of IFX levels by POCTs compared to ELISAs, resulting in significant variations in the classification of individuals in the supratherapeutic range of IFX [27,28,29]. When patients had a low infliximab trough level, all methods were relatively comparable; however, when the infliximab level was greater than 3 mg/L, there was a lack of concordance. Furthermore, some POC techniques overestimate the trough level compared to other techniques, which can create difficulties in optimising treatments for patients [30,31].

Concerning ADM trough levels, the results show a great concordance between the ELISA-based technique and the POCT, with a low bias. Additionally, the level categorisation based on the therapeutic ADM ranges revealed few differences (10%) between the two assays. These results demonstrated that the two kits for quantitative determination of ADA were comparable and suitable for clinical use. However, the transition from one assay to another requires evaluation at the routine laboratory level to ensure that clinicians can interpret results for use in patient management, and, where possible, longitudinal measurements should be performed on the same kit from the same brand.

Regarding the comparison of the detection of ADAs to IFX and ADM, our study found an almost perfect agreement between the two assays in ADA detection, in which over 90% of the results are concordant for both comparisons. The sensitivity of the two assays to detect antibodies was comparable, with a slight advantage for the RIDASCREEN^®^ one in detecting low ADA levels (the minimal detectable concentration is 0.06 ng/mL). However, these low levels of ADAs would have minimal effect on clinical decisions; as a result, changing the therapy is not required [22]. There was just one measurement with a false-negative result (in ADAs to ADM detection) from the CHORUS assay that, in contrast, was quite high with the RIDASCREEN^®^ one. Due to overexposure, the value could be mistakenly considered negative since it appears to be oversaturated. The detection of conventional and clinically significant levels of ADAs without producing false positives or negatives was the major objective of this comparison test. Only for ADAs to IFX were cut-off values established to discriminate between undetectable, low, and high ADA concentrations in the literature [12,23]. Importantly, these cut-offs are highly assay dependent and cannot be interchanged between different assays [23]. Consequently, it is challenging to compare the outcomes of these two kits using cut-offs. We noticed that not all manufacturers provide a cut-off or therapeutic window for the interpretation of the anti-TNFα concentrations or ADA levels. However, these target values are crucial due to how they could influence clinical decision making and result in significant therapeutic changes.

The study’s limitations include the relatively small sample size and the fact that proactive TDM was applied to the majority of the samples that were collected, which limited the evaluation of clinically significant differences at lower drug concentrations.

## 5. Conclusions

Quantitative and qualitative differences between the two assays used to measure anti-TNFα inhibitors (mainly IFX) were identified in the present study. Despite the significant statistical association, these tests are not fully interchangeable. Particularly, the systematic biases in IFX high levels and the discrepancies found in concentration ranges based on the therapeutic window demonstrate that subsequent therapeutic choices could differ based on the assay. Considering that the clinical impact of these discrepancies has not been sufficiently investigated, and until commercial tests are extensively cross-validated and standardised, patients should preferably be monitored and managed with the same TDM assay (ideally from the same manufacturer) over time.

## Figures and Tables

**Figure 1 pharmaceutics-15-01834-f001:**
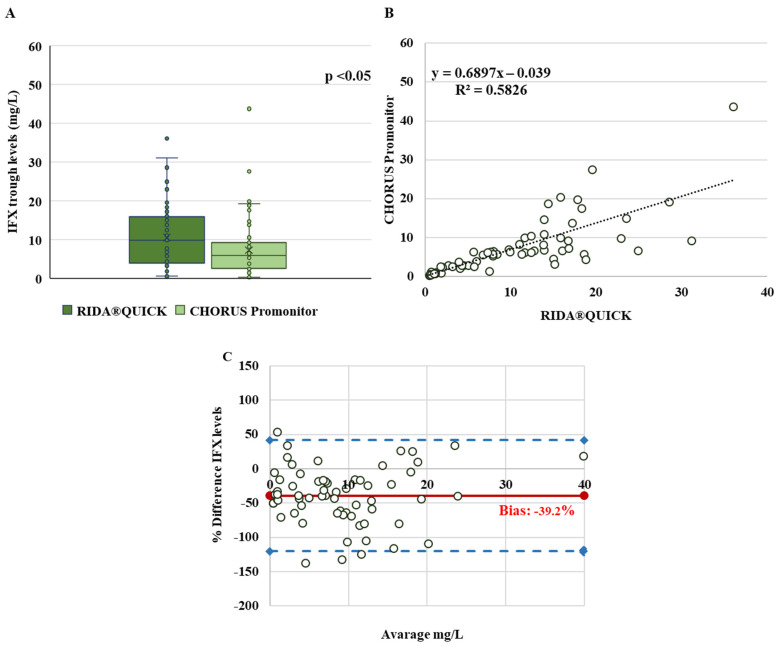
Comparison between RIDA^®^QUICK IFX Monitoring and Chorus Promonitor IFX assays. (**A**) Box plot comparing IFX plasma concentrations (mg/L), the horizontal line in the middle represents the median; (**B**) linear regression of IFX levels; (**C**) Bland–Altman plots: the difference between the two measurements (mg/L) is plotted on the y-axis and the average of the two measurements (mg/L) on the x-axis. Blue dashed lines represent the 95% limits of agreement of the bias. Bias (in solid red) is then calculated according to the Bland–Altman method.

**Figure 2 pharmaceutics-15-01834-f002:**
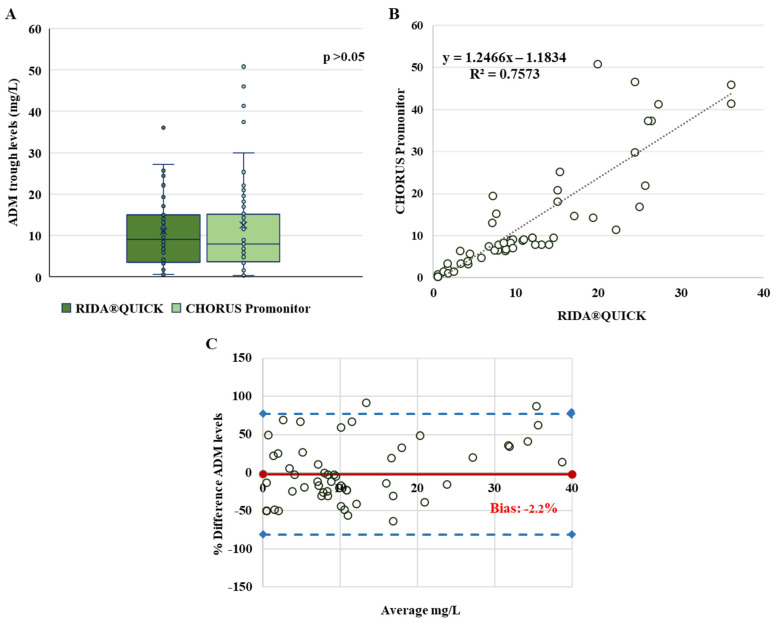
Comparison between the RIDA^®^QUICK Monitoring ADM and Chorus Promonitor ADL assays. (**A**) Box plot comparing ADM plasma concentrations (mg/L), the horizontal line in the middle represents the median; (**B**) linear regression of ADM levels; (**C**) Bland–Altman plots: the difference between the two measurements (mg/L) is plotted on the y-axis and the average of the two measurements (mg/L) on the x-axis. Blue dashed lines represent the 95% limits of agreement of the bias. Bias (in solid red) is then calculated according to the Bland–Altman method.

**Table 1 pharmaceutics-15-01834-t001:** Comparison of IFX concentration results obtained by the RIDA^®^QUICK and CHORUS Promonitor assays stratified by therapeutic interval.

Groups	N° of Patients (%)		Kappa Statistics		
IFX concentrations	RIDA^®^QUICK N = 65	CHORUS Promonitor N = 65	K	SE	CI 95%
<3 mg/L	15 (23)	22 (33.8)	0.410	0.078	0.257–0.563
≥3 to ≤7 mg/L	10 (15.5)	23 (35.4)			
>7 mg/L	40 (61.5)	20 (30.8)			

K: Cohen’s kappa coefficient, SE: the standard error(s) of the kappa coefficient, CI 95%: 95% confidence interval.

**Table 2 pharmaceutics-15-01834-t002:** Comparison of ADM concentration results obtained by the RIDA^®^QUICK and CHORUS Promonitor assays stratified by therapeutic interval.

Groups	N° of Patients (%)		Kappa Statistics		
ADM concentrations	RIDA^®^QUICK N = 58	CHORUS Promonitor N = 58	K	SE	CI 95%
<5 mg/L	18 (31)	17 (29.3)	0.66	0.082	0.503–0.824
≥5 to ≤10 mg/mL	16 (27.6)	22 (38)			
>10 mg/L	24 (41.4)	19 (32.7)			

K: Cohen’s kappa coefficient, SE: the standard error(s) of the kappa coefficient, CI 95%: 95% confidence interval.

**Table 3 pharmaceutics-15-01834-t003:** Qualitative agreement between the two assays for ADAs to IFX and ADM detection in plasma samples.

Groups		N° of Samples (%)			Kappa Statistics		
Anti-IFX Antibodies		RIDASCREEN^®^ (ng/mL)			K	SE	CI 95%
		Positive	Negative	Total			
CHORUS Promonitor (AU/mL)	Positive	25 (78.1)	0 (0)	25 (78.1)	0.904	0.094	0.719–1.000
	Negative	1 (3.1)	6 (18.8)	7 (21.9)			
	Total	26 (81.2)	6 (18.8)	32 (100)			
Anti-ADM Antibodies		RIDASCREEN^®^ (ng/mL)			K	SE	CI 95%
		Positive	Negative	Total			
CHORUS Promonitor (AU/mL)	Positive	13 (56.5)	0 (0)	13 (56.5)	0.819	0.120	0.583–1.000
	Negative	2 (8.7)	8 (34.8)	10 (43.5)			
	Total	15 (65.2)	8 (34.8)	23 (100)			

K: Cohen’s kappa coefficient, SE: the standard error(s) of the kappa coefficient, CI 95%: 95% confidence interval.

## Data Availability

Not applicable.

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
