# Peer review of "Therapeutic Drug Monitoring of Anti-TNFα Inhibitors: A Matter of Cut-Off Ranges"

_pharmaceutics, 2023, doi:10.3390/pharmaceutics15071834_

Round 1
Reviewer 1 Report
The authors have presented a clinically relevant subject relating to the relevance of therapeutic drug monitoring of anti-TNFa inhibitors in inflammatory bowel disease affected patients. The specific case chosen for the study is the investigation of comparator aspects between automated immunoassay ELISA-based (CHORUS Promonitor) and the lateral flow assay (RIDA® QUICK) for quantifying infliximab and adalimumab levels in plasma of such patients. The qualitative and quantitative interchangeability between two assays does not exist according to the study findings.
The subject of the article is indeed relevant considering its clinical relevance and utility. Authors have done a good job in designing the study appropriately and considering all the comparator aspects between two assays. The findings have been appropriately documented and well explained. However, the writing style could have been improved so as to easily convey the findings to the readers. The current paragraphs are not very organized and becomes distracting at many instances due to the mix up of sentences from different contents. Some suggestions which can be considered during revision are given below:
1. Section 1- background, lines 40-42: “although few studies….after induction” seem to be confusing- please reframe this statement appropriately.
2. Section 1, paragraph 2, lines 49-74: the paragraph seems to be highly unorganized with mix up continuous sentences relating to different contents and therefore it seemed to be distracting. It would be better if authors can split up further to paragraphs as needed or organize the sentences in the form of a table or a different format for better understanding.
3. Section 1- background, lines 72-74: “conventional immunoassays.. lack the ability to detect only free ADAs”- What ability does it specifically lack?
4. Section 2.1, lines 105-106: “The main assay differences regard the units and immune complex revelation steps.”- this statement needs to be better explained.
5. Section 2.3- please provide more specific details regarding the patient including age, gender etc. and the rationale of selection criteria.
6. Section 3.1, lines 181-183: “Since the RIDASCREEN® assay exhibited higher IFX concentrations than the CHORUS Promonitor one, the classification range for all these samples was changed and they were consequently shifted to a higher category.”- what was the change implemented for classification range? Please provide additional information to explain clearly.
7. Section 3.3, line 239: how would you justify using such dilution value for this instance? Is it recommended by the manufacturer?
8. Section 4, line 285: ‘low ADAs levels’ – please mention the suitable range of values here.
9. Section 4, line 286: Why would the clinical decision stay unaffected by low ADAs levels? This needs to be better explained with literature support.
10. Addition of more detailed consolidating statements to the section on conclusions is recommended.
Minor editing required.
Author Response
We thank Reviewer 1 for his/her thorough review, and we greatly appreciate the comments and suggestions, to which we respond as follows:
- Section 1- background, lines 59-61 (ex-lines 40-42) - We have modified the text as follows: “Increasing evidence in children supports the update of European guidelines on optimising the treatment of IBD by early proactive TDM followed by dose optimisation in patients on anti-TNF agents”.
- Section 1, paragraph 2, lines 84-87 (ex-lines 49-74) - We have modified the text to include all observations made in his/her comment as follows: “The main benefits of an ELISA assay are its user friendliness and the commercially available ready-to-use kits. The fluid phase systems (RIA, HMSA) are well known for their ability to identify low affinity and monospecific ADAs but are indeed less frequently implemented in the clinical practice since they are more demanding techniques. These methods are time-consuming and require sophisticated equipment, reagents, and technical expertise. The development of a technique based on rapid point-of-care tests (POCT) in lateral flow immunochromatography, however, has simplified the methodology to measure plasma IFX and ADM concentrations. This provides advantages such as the possibility to test individual sample, the comparatively simple operation, and the quick turnaround for results (15–20 minutes). The IFX and ADM trough levels measurement has been compared in numerous studies showing, overall, good correlations between the methods (14-17). However, several studies reported differences in the absolute drug concentrations, and this can impact on the clinical practice when thresholds or ranges of concentration are the target to reach by TDM (15-17).”
- Section 1- background, lines 98-100 (ex-lines 72-74) - We have modified the text to better explain: “Conventional immunoassays, such ELISA, are routinely used for ADAs screening, although they have a low drug tolerance and cannot distinguish neutralizing and non-neutralizing ADAs”.
- Section 2.1, lines 128-132 (ex-lines 105-106) - We revised the sentence to better clarify: “The main assay differences regard the units and immune complex revelation steps: RIDASCREEN® expresses ADAs results in ng/ml whereas CHORUS Promonitor gives results in arbitrary units; the other difference is the revelation strategy which is a two-step biotin/streptavidin revelation for RIDASCREEN®, as opposed to CHORUS Promonitor, which is a one-step procedure.”
- Section 2.3 - We thank the Reviewer for this comment. We compared methods by determining patient TDM data without evaluating patient clinical data.
- Section 3.1, lines 215-217 (ex-lines 181-183) - The sentence has been revised to adequately explain the concept: “Since the RIDASCREEN® assay exhibited higher IFX concentrations than the CHORUS Promonitor one, the classification range for all these samples was different between the two assays, with 30% of these patients shifting to a higher category.”
- Section 3.3, line 272-273 (ex-lines 239) - In the text, we have inserted the comment to better clarify, as follows:” It was possible to measure ADAs in this patient with the CHORUS Promonitor test after diluting the sample up to 1:10.000 with a result of 20.500 AU/mL.”
- Section 4, line 319-321 (ex-lines 285) - We have modified the text to better explain: “The sensitivity of the two assays to detect antibodies was comparable, with a slight advantage for the RIDASCREEN® one in detecting low ADAs levels (the minimal detectable concentration is 0.06 ng/ml).”
- Section 4, line 321-322 (ex-lines 286) - In the text, we have inserted the comment to better clarify, as follows: “However, these low levels of ADAs would have minimal effect on clinical decision; as a result, it doesn't require changing the therapy (22).”
- Conclusion- We thank the reviewer for this comment. We have modified the text to include all observations made in his/her comment as follows: “Particularly, the systematic biases in IFX high levels and the discrepancies found in concentration ranges based on the therapeutic window, demonstrated that subsequent therapeutic choices could differ based on the assay.”

Reviewer 2 Report
Therapeutic drug monitoring (TDM) is a useful tool for optimizing the use of anti-TNFα inhibitors in patients with inflammatory bowel diseases (IBD). Authors evaluated the performance, interchangeability, and agreement between an automated immunoassay ELISA-based (CHORUS Promonitor) and the lateral flow assay (RIDA®QUICK) for the quantification of infliximab (IFX, n=65) and adalimumab (ADM, n=58) plasma levels in IBD patients. 26 samples for IFX and 16 samples for ADM, that tested positively for the presence of ADAs, were also used. Despite the significant,statistical association, these tests are not fully interchangeable. Considering the clinical,impact of these discrepancies has not been sufficiently examined, and until commercial tests are completely cross-validated and standardised. Authors concluded that the patients should preferably be monitored and managed with the same TDM assay
The results are well supported by mathematical statistical analyzes as well as graphical representations.
1. In the introduction, the authors could give a schematic comparison of different analytical methods.
2. In the supporting material, they could describe in more detail why and when they use certain statistical tests-methods, for example Kappa statistics, this would make it much easier to follow the manuscript, especially for PhD students.
Author Response
We thank the Reviewer 2 for his/her thorough review, and we greatly appreciate the comments and suggestions, to which we respond here below:
- Introduction- Lines 83-95 We have modified the text to to better clarify, as follows: “The main benefits of an ELISA assay are its user friendliness and the commercially available ready-to-use kits. The fluid phase systems (RIA, HMSA) are well known for their ability to identify low affinity and monospecific ADAs but are indeed less frequently implemented in clinical practice since they are more demanding techniques. These methods are time-consuming and require sophisticated equipment, reagents, and technical expertise. The development of a technique based on rapid point-of-care tests (POCT) in lateral flow immunochromatography, however, has simplified the methodology to measure plasma IFX and ADM concentrations. This provides advantages such as the possibility to test individual sample, the comparatively simple operation, and the quick turnaround for results (15–20 minutes). The IFX and ADM trough levels measurement has been compared in numerous studies, showing overall good correlations between the methods (14-17). However, several studies reported differences in the absolute drug concentrations, and this can impact on the clinical practice when thresholds or ranges of concentration are the target to reach by TDM (15-17).”
- Supporting material- We have added supporting material to describe statistical methods in more detail.

Reviewer 3 Report
The paper reports the comparison between two commercially available kits for monitoring the plasma levels of MAB and ADAs in a hospital setting. It would have been expected that the kits should have performed similarly, however the differences seen for IFX plasma values are surprising, especially since these kits should have been validated in patient samples by the manufacturer against other accurate techniques. The experimental methods for analysis are satisfactory. I have a few comments and will appreciate if the authors address them.
Major comments
11) Please define and discuss ADAs in brief in the manuscript.
22) Comparing the two methods, is it correct to say that the Elisa based methods is more accurate? Or is it a manufacturer related assay design flaw leading to the differences in IFX levels, but not for ADM? Can these be related to operator variability?
33) TDM is vital for adjusting drug doses and successful therapy, can the authors recommend some proactive steps for doctors and other healthcare professionals to identify and minimize the chances of subtherapeutic dosing in the paper?
The quality of English is fine, but I would still recommend proofreading the paper.
Author Response
We thank the Reviewer 3 for his/her thorough review, and we greatly appreciate the comments and suggestions, to which we respond here below:
1) In the introduction paragraph (lines 67-72), we have modified the text to include all observations made in his/her comment as follows: “Up to 73% of patients receiving IFX and up to 35% of patients receiving ADM report the formation of persistent ADAs (9). About one-third of these patients usually experience a loss of response because of developing ADAs, which usually occurs within 12 months after the start of treatment (9). Therefore, to improve the management of patients with IBD, anti-TNFa drugs trough levels and ADAs determination are commonly carried out (so-called proactive TDM) (9-10).”
2) We thank the Reviewer for this comment. In this manuscript, we would like to emphasize that the cut-off values developed for the ELISA assays may not be suitable for the use of the POCT. In particular, the greatest discrepancies were found in the high levels of IFX (30%) resulting in poor agreement according to the Kappa result (0.41). These discrepancies can be explained by operator or inter-assay variability. These assays are based on automated or semi semi-automated processes which reduced operator-specific influence on test results. In addition, both kits were compared to the corresponding kits (predicate devices) and with International Standard for IFX and ADM to assess the accuracy and linearity. Further study with long-term follow-up using these methods may help comprehend and explain these variations.
3) We thank the Reviewer for this observation. In the introduction paragraph (lines 72-76), we have inserted a comment as follows: “Numerous exposure-outcome relationship data from prospective studies and post-hoc analyses of randomized controlled trials (RCTs) have shown the benefits of proactive TDM over empiric or reactive dosing (1,11-14). Proactive TDM can also have an important role when de-escalating therapy, proving to be more cost-effective (1,11-14).”

Reviewer 4 Report
The authors evaluated the performance, interchangeability, and agreement between an automated 19 immunoassay ELISA-based (CHORUS Promonitor) and the lateral flow assay (RIDA® QUICK) for 20 the quantification of infliximab (IFX, n=65) and adalimumab (ADM, n=58) plasma levels in IBD 21 patients.
the following comments may enhance the manuscript quality:
- General revision of the font size of titles and subtitles.
- Add inclusion and exclusion criteria of sample selection (In table form).
- Revise (<3 mg/L, ≥3 to 7 mg/L and <7 mg/L for IFX concentrations and <5 145 mg/L, ≥5 to 10 mg/L and <10 mg/L for ADM concentrations, according to the 146 manufacturer's instruction)......I think more than 10 mg/L for ADM.
- Mach the sample size in the abstract with results, 3.1, 3.2, and 3.3.
- Discussion can be improved to reflect the valuable results of the study.
- Conclusion can be improved
Moderate editing of English language required
Author Response
We thank the Reviewer 4 for his/her thorough review, and we greatly appreciate the comments and suggestions.
1) The font sizes for titles and subtitles have been revised.
2) Lines 152-155, we have presented in the text the inclusion and exclusion criteria as follow: “All patients with an established diagnosis of Crohn’s disease (CD) or ulcerative colitis (UC) were included. Plasma samples were collected at trough level both during the induction and maintenance phases. Samples were excluded if there wasn't sufficient plasma to run them through both assays and if they were homolyzed. In addition, patients were excluded if data on IFX or ADM dosing or levels were missing.”
3) We have reported in this paper the therapeutic windows reported by the kit instructions which are:
- for IFX: a target therapeutic trough concentration window of 3-7 µg/ml is recommended, following the TAXIT algorithm (Vande Casteele N, et al., Gastroenterology 2015, ref. n°11).
- for ADM, a target therapeutic trough concentration window of 5-10 µg/ml has been recommended, following the literature (Papamichael et al., Frontline Gastroenterol 2016, ref. n°10).
4) We thank the reviewer for this comment. We have modified the text to include all observations made in his/her comment as follows:
- Lines 34-36. “the quantification of infliximab (IFX, n=65) and adalimumab (ADM, n=58) plasma levels in IBD patients. 32 samples for IFX and 23 samples for ADM, that tested positively for the presence of ADAs, were also used.”
- Lines 156-160. “A total of 123 (65 for IFX and 58 for ADM) plasma samples from 123 IBD patients treated with anti-TNFα inhibitors were tested for trough concentration measurement using RIDA®QUICK Monitoring and the CHORUS Promonitor test. Forty-two samples were tested positive for the presence of ADAs (26 for IFX and 16 for ADM) by the RIDASCREEN® assays, which were also analysed with the CHORUS Promonitor tests.”
- Lines 179-180. “The IFX plasma concentrations, measured in 65 IBD patients for each method, were first checked for normality, and data from the CHORUS Promonitor assay showing a non-normal distribution.”
- Lines 225. “The plasma ADM trough levels of 58 IBD patients were measured.”
5) We thank the reviewer for this comment. We have modified the text to include all observations made in his/her comment as follows:
- Lines 303-309. “Previous method comparison studies have demonstrated an overestimation of IFX levels by POCT compared to ELISA assays, resulting in significant variations in the classification of individuals in the supratherapeutic range of IFX (27-29). When patients had a low infliximab trough level, all methods were relatively comparable; however, when the infliximab level was greater than 3 mg/L, there was a lack of concordance. Furthermore, some POC techniques overestimate the trough level compared to other techniques, which can create difficulties in optimizing patients (30).”
6) We thank the reviewer for this comment; we have changed the text to include all his/her observations, as follows:
- Lines 342-344: a comment has been added in the text to better explain:” Particularly, the systematic biases in IFX high levels and the discrepancies found in concentration ranges based on the therapeutic window, demonstrated that subsequent therapeutic choices could differ based on the assay.”
Round 2
Reviewer 3 Report
I thank the authors for their response to my comments.
Reviewer 4 Report
The authors replied to all comments
Good quality of English Language